# Translation and psychometric evaluation of chronic illness anticipated stigma scale (CIASS) among patients in Ethiopia

**Mohammed Hassen Salih**[1]⊗*, **Lena Wettergren**[2]⊗, **Helena Lindgren**[3]⊗, **Kerstin Erlandsson**[4]⊗, **Hussen Mekonen**[5]‡, **Lemma Derseh**[6]‡

1 School of Nursing, College of Medicine and Health Sciences, University of Gondar, Gondar, Ethiopia, 2 Department of Public Health and Caring Sciences, Uppsala University and Department of Women's and Children's Health, Karolinska Institute, Solna, Sweden, 3 Department of Women's and Children's Health, Karolinska Institute, Solna, Sweden, 4 School of Education, Health and Social Studies, Dalarna University and Department of Women's and Children's Health, Division of Reproductive Health, Karolinska Institute, Solna, Sweden, 5 School of Nursing and Midwifery, College of Nursing and Midwifery, Addis Ababa University, Addis Ababa, Ethiopia, 6 College of Medicine and Health Sciences, Institute of Public Health, University of Gondar, Gondar, Ethiopia

⊗ These authors contributed equally to this work.
‡ HM and LD also contributed equally to this work.
* muhenet@gmail.com

**Data Availability Statement:** All relevant data is available on Mendeley at DOI 10.17632/

## Abstract

### Background

Stigma is common among patients with chronic illnesses. It affects the delivery of healthcare for not addressing the psychological components and may interfere with the patient's attendance to necessary health care services. Therefore, a valid and reliable instrument to measure anticipated stigma related to chronic illness is vital to inform possible interventions. This study aimed to translate the Chronic Illness Anticipated Stigma Scale (CIASS) into the Amharic language and evaluate its psychometric properties in Ethiopia.

### Methods

The CIASS was translated into Amharic language using standard procedures. The Amharic version was completed by 173 patients (response rate 96%) with chronic illness from three referral hospitals in the Amhara region. Internal consistency was examined through Cronbach's alpha. Construct validity was evaluated by confirmatory factor analysis and convergent validity by using a Pearson correlation of P-value less than or equal to 0.05.

### Results

The internal consistency was estimated at Cronbach alpha of 0.92. By using a structural equation model, and modification indices a model fitness testing was run and shows a root mean squared error of approximation 0.049 (90% CI, 0.012–0.075). The structural validity results in 78.8% of confirmatory factor analysis showed from the extraction of the three-dimension (components). Validity tests for convergent by using Pearson correlation positively correlated with common mental distress and negatively correlated with quality of life–BREF, and the construct validity shows a good valid tool to CIASS.

gvrk8njd76.1. Also attached here in the excel format.

**Funding:** The author(s) received no specific funding for this work.

**Competing interests:** The authors have declared that no competing interests exist.

**Abbreviations:** CFI, Comparative Fit Index; CIASS, Chronic Illness Anticipated Stigma Scale; DRH, Dessie Referral Hospital; EFA, Exploratory Factor Analysis; FACIT, Functional Assessment Of Chronic Illness Therapy; FHRH, Felege Hiowt Referral Hospital; HCW, Health Care Workers; HIV/AIDS, Human Immunodeficiency Viruse /Acquired Immunodeficiency Syndrome; K10, Kessler psychological distress scale; RMSEA, Root Mean Squared Error Of Approximation; SD, Standard Deviation; SEM, Structural Equation Model; TLI, Tucker-Lewis Index; UGCRH, University Of Gondar Comprehensive And Referral Hospital; WHOQOL-BREF, World Health Organization Quality Of Life–Bref.

## Conclusion

The Amharic language version of the chronic illness anticipated stigma scale shows a satisfactory level of reliability and validity on different psychometric measures of assessment. The tool may be useful for future researchers and patients with chronic illness in the Amharic-speaking population. Moreover, it will be used to see the psychological burden related to chronic illness and for comparison among international population groups.

## Introduction

Stigma is an attribute that labels a person in discriminatory ways. Erving Goffman (1963) classically defined stigma as an "attribute that is deeply discrediting". Stigma is the sign that indicates something out of normal about the person [1]. In 2001, Link and Phelan expanded the stigma model to include loss of social status in addition to discrimination experiences [2]. It is shifting from the 'kingdom of the well' to the 'kingdom of the sick [3].

There are three types of stigma. Namely, enacted, anticipated, and perceived stigma. Anticipated stigma is the perception/belief of the individual that prejudice, discrimination, and stereotyping will likely occur in the future [4]. In other words, it is an anticipation of the enacted stigma [5]. In the case of internalized stigma, the affected individual has internalized negative stereotypes or negative attitudes, may feel ashamed or guilty because of their condition [6–10].

Anticipated stigma has been associated with undesirable health outcomes such as poor adherence to their prescribed medication, loss of attachment for their follow-up, and poor progress to their medical recovery [11, 12]. People living with chronic illnesses who had stigma experienced by healthcare workers will anticipate greater stigma from healthcare workers and in turn, it will lead to less access to healthcare [13].

Different types of stigma measurements have been developed; the stigma scale for chronic illnesses 8 item version (SSCI-8) to assess stigma in patients with neurological disorders [14], the stigma of mental illness [15], the discrimination and stigma scale (DISC-12) [16], the Internalized Stigma of Mental illness Inventory- 9 Version (ISMI-9) [17] and the CIASS-12 developed by Earnshaw and coworkers [4].

Stigma may be experienced and affect a person in different forms (physical and psychological) and we will now describe the translation process and psychometric evaluation of the Amharic language version of the chronic illness anticipated stigma scale. The translation of appropriate tools related to anticipated stigma applicable for patients with chronic illness as important steps to assess anticipated stigma among patients with chronic illness.

The CIASS was developed to measure anticipated stigma (i.e., expectations of prejudice, stereotyping, and discrimination) among people living with chronic illnesses. It is expected to measure the experience of discrimination, prejudice, and stereotyping from others in the future [4].

Measures of anticipated stigma are very important if they take specific sources of stigma into account. However, most stigma scales and tools ignore the sources of the stigma only asking about the extent to which they anticipate stigma from others in general, it is experienced rather than from specific groups; such as family members, employers, or healthcare workers. The CIASS was developed and tested psychometrically in different communities and languages, among them, the English original version, Persian version, and the Spanish Version where recent one [4, 18, 19].

At the level of our search, there is no published psychometrically validated scale to measure anticipated stigma associated with chronic illness in the Amharic-speaking population in Ethiopia. Therefore, translating and evaluating the psychometric properties of the Amharic version of the anticipated stigma scale will be important for other researchers for continuous management and follow-up among patients.

This study was aimed to translate the Chronic Illness Anticipated Stigma Scale (CIASS) into the Amharic language and evaluate its psychometric properties in Ethiopia.

## Methods and materials

### Study area, and period

The study was conducted in Amhara Region from December 2020 to 15 January 2021. The region has 5 Referral Hospitals and among those, we selected three referral hospitals for the study.

### Population, sample size, and sampling procedure

Patients who were on follow-up visits with a diagnosis of chronic illness in Amhara Regional State Referral hospitals constituted the population to be studied. Adult patients whose age was 18 or above, who were at least for one-month follow-up in chronic illness clinic, or correctly to respond to the cognitive criteria for mental illness clinic [20] were included in the study. However, acutely sick and admitted patients were excluded.

The terms *chronic illness* and *chronic disease* are often used interchangeably in clinical areas and health services programs. However, they convey different meanings that require clarification [1]. *Chronic illness* is the personal experience of living with the condition that often accompanies chronic disease [2, 3]. *Chronic disease* is defined based on the biomedical disease classification and it requires continuous management over a long period. In our paper, we included cardiovascular and respiratory disorders, cancer, diabetes, asthma, long-term viral conditions such as Hepatitis C, communicable diseases, for example, HIV/AIDS, and mental health disorders [4–6].

Based on different scholars, the sample size required for tool validation ranges from 5 to 20 patients per item [21]. The CIASS encompasses 12 items and by considering 15 patients per item, a total of 180 patients attending chronic illness clinics were approached.

Patients accepting participation were interviewed by a research assistant with a Master's degree and BSc in health (Nursing).

The study participants were selected based on the weekly patient load in the chronic illness follow-up clinic in the three Amhara Region Referral Hospitals (University of Gondar Comprehensive Specialized Referral Hospital, Felege Hiwot Referral Hospital, Dessie Referral Hospital). Patient identification number daily list was used as a sampling frame. With proportional allocation to each hospital, 79, 61, and 40 patients, respectively, were included in the study. Proportional allocation for each unit was also used to reach the desired sample from each chronic illness follow-up clinic but not less than 5 patients in each unit.

Before the main data collection, 13 patients living with chronic illness were pretested for cognitive debriefing. For this interview patients were purposively selected considering their socio-demographic and clinical conditions.

### Variables and measures

The CIASS is a measure developed by Earnshaw and colleagues (2013). It includes 12 items divided into three subscales evaluating the extent to which the patients anticipate stigma from

friends and family members, work colleagues, and healthcare workers. In family and friends sub item it contains four questions like "a friend or family member will be angry with you". In the work colleagues' sub-item, it included four questions like "your employer will not promote you". And in the healthcare providers/workers sub-item, included four questions like "a health care worker will be frustrated with you". Participant responses are indicated on a Likert-type scale ranging from 1 (very unlikely) to 5 (very likely) [4]. The average anticipated stigma score is calculated by adding up all items and averaging with the total number of items.

For the assessment of convergent validity, the Kessler psychological distress scale (K10) and the World Health Organization Quality Of Life–Bref (WHOQL-BREF) were used. Patients experiencing stigma are prone to increase psychological distress, such as depression [22]. Additionally, previous studies observed that quality of life was associated with stigma [23–25].

Psychological distress was measured by using the Kessler 10 psychological distress scales (K10) [26]. This instrument has 10 items each asking the respondent how often she/he experienced symptoms during the past 30 days and containing 5-point Likert scales (1 = never, 2 = a small part of the time, 3 = some of the time, 4 = most of the times, 5 = all of the time). The Kessler-10 scale has been validated in Ethiopia and used extensively [27, 28].

WHOQOL- BREF is a 26—item instrument consisting of four domains: physical health (7 items), psychological health (6 items), social relationships (3 items), and environmental health (8 items), and two items not included in any of the domains are an overall perception of quality of life (QOL) and general health. Each of these items was scored from 1 to 5 on a response scale, which is agreed on as a five-point Likert scale. The domain scores were calculated according to the manual for WHOBOL-BREF [29] and it was validated in the Amharic version on type 2 diabetes patients [30].

## Linguistic validation

The CIASS was translated into Amharic language using the forward-backward process of the Functional Assessment of Chronic Illness Therapy (FACIT) translation methodology [31].

The translation process included several steps: forward translation, discussion with translators, back-ward translation, expert discussion, and cognitive test on purposively selected patients.

The original English version of the instrument was translated to the Amharic language version by using forward translation to ensure semantic equivalence by two native Amharic-speaking original translators.

The two translations of the Amharic version were discussed with the main investigator (PI) to create an initial Amharic version. The initial Amharic version was then back-translated into English by two language experts (the experts are from the University of Gondar English language department).

A one-day session was held to examine the forward translated and back-translated versions by behavioral and social related experts, mental health specialists, and chronic nurse specialists with extensive knowledge of the chronic illness patient assessment for more than 16 years. They examine all of the proceeding steps and selected the most appropriate translation for each item or provided alternate translations. The overall discussion and meeting with experts were led by the primary investigator.

To determine the readability of the language used, cultural context, and to ask for their feedback on the comprehensiveness of items cognitive debriefing was conducted with a convenience sample of 13 participants at the University Of Gondar Comprehensive Specialized Hospital. During cognitive debriefing, patients were asked to say something on the overall comprehensiveness of the tool, and some modifications in some words were done. We added

**Table 1. The Linguistic validation process for the Amharic version of anticipated stigma scale among chronic illness patients in Amhara Region Referral Hospitals.**

| | |
|---|---|
| **Original instrument translation to Amharic** | **Forward Translator (FT1)**—He has familiarity with the literature on health-related psychology and epidemiology masters (Fluent in the Amharic, good understanding of English). |
| | **Forward Translator (FT2):**—Fluent in the Amharic and Good understanding of English, but without health-related background and not informed about the tool. |
| **A Synthesized translated version** | **The PI and** the two forward translators assessed the two forward translated versions to identify major discrepancies. |
| **Back translations** | **Back translator BT1**- Language and subject expert, a good understanding of Amharic (He has familiarity with the literature on health-related, Ph. D in nursing student) |
| | **Back translator BT2**—Language expert in English and a good understanding of Amharic. |
| **Expert committee meeting for one day** | Behavioral and social related experts, mental health specialists, and Chronic nurse specialists who had extensive knowledge and experience of chronic illness patient care. |
| | The patient-relevant score for each item and total comprehensiveness was assessed. |
| **Cognitive interviews (pilot interview)** | Thirteen (13) patients were purposively selected from the chronic illness clinic at the University of Gondar Comprehensive Specialized and Referral Hospital. |
| **Finalized the instrument** | The principal investigator evaluated all over the content. |

Key:- BT = Back Translator, FT = Forward translator, PI = Principal Investigator.

During cognitive debriefing 13 patients were interviewed by purposive sampling technique, 93.6% of the participant said it was completely relevant for the chronic illness. Their mean and SD were 2.51 & 0.5 respectively.

an explanation on the working place for those who had no experience of being employed, like in the Ethiopian context, the experiences of the farmers. In conclusion, there were no major differences from the original English language version (Table 1).

## Data analysis

Validity was assessed by construct and convergent validity. Construct validity was examined by Confirmatory factor analysis (CFA) by using the principal component analysis and the vari-max rotation method was used. Also, scale and item analysis, and Pearson's correlations were performed.

Convergent validity was examined by assessing correlations between the CIASS and the instruments WHOQOL BREF and the K10. Based on different scholars' quality of life is negatively correlated to stigma and positively correlated to psychological distress.

Reliability was assessed by internal consistency. The internal consistency was calculated by using the Cronbach's-α coefficient. Cronbach's-α of 0.70 or greater is considered to indicate acceptable reliability [32].

Data entry at epi info V7 and exported to IBM SPSS Version 25 and STATA Version 14 was used for different psychometric test statistical analyses. All data, except identification and some socio-demographic variables, was made available at Mendeley data depository [33].

## Data quality

To maintain the quality of data collection instruments' discussion was held with 14 data collectors and three supervisors to have a common understanding. Before the actual data collection, two-day training was given for the data collectors and supervisors.

Before the main data collection 36 patients were done for inter-rater reliability for 14 data collectors. Their result was 0.875 average measure intraclass correlation coefficient with 95% Confidence interval (0.775–0.936).

During data collection, the assigned supervisors followed each activity and gave direct support. Responses for more than 5% of the patients who had not answered, i.e. missing values of two or more questions from 12 items were excluded. But for those less than 5% correction of responses to the average score was done. The five-point Likert scale of the anticipated stigma scale was evaluated through analysis of whether all response alternatives, 1–5 were used for all items. For the total scale floor and ceiling effects were calculated [34].

## Ethical consideration

Permission was obtained from the tool developer (Dr. Valerie Earnshaw). The ethical clearance letter was received from the University of Gondar ethical review board and written consent was obtained from each study site. Because most of the participants are illiterate, the University of Gondar ethical review board approved to use of either verbal or written consent with letter reference number V/P/RCS/05/89/2020 on September 17th, 2020.

Before each data collection, the signature of each patient or verbal consent was secured on the first page of the consent paper. Information related to the aim of the study, voluntariness, and benefits in participating in this study was clearly stated before data collection.

The principal investigator, supervisors, and data collectors ensured the privacy and confidentiality of information throughout the study process. Information sheets and consent forms were placed in a secure place.

Confidentiality was secured at all stages and that data would be presented only for educational purposes.

## Results

### Socio-demographic characteristics of the participant

From the total 180 patients interviewed, 173 were included in the psychometric evaluation (response rate of 96%). One hundred four (60%) of the participants were male. The mean (standard deviation) age was 44 (13.8). The majority (78%) of the participants were urban residents, more than two-thirds (78%) were orthodox. Almost two-thirds (63%) of them were married. Regarding working conditions, more than two-thirds (71%) of them were working in a private or non-governmental institution. One hundred forty (80.9%) of them got information from mass media on facts related to their health, from them around two-thirds (68.9%) had got information from television (Table 2).

### Clinical characteristics of the participants

More than two-thirds (72.8%) of them have fear of contagion in the hospital and the vast majority (92%) reported they attended regular follow-up.

Thirty-four patients were diagnosed with mental illness, 34 with HIV/AIDS, and 30 patients with Diabetic Mellitus. Two-thirds (67.6%) of them had a single diagnosis. The mean (SD) for the year since diagnosis was 6.9 (5.3) (Table 3).

### Reliability tests

**Internal consistency.** The Cronbach's-α coefficient for the reliability of the CIASS was 0.92. Cronbach's-α value for anticipated stigma to family and friend dimension is 0.93, to work colleagues 0.91 and to health care workers 0.88. Mean (SD) for chronic illness anticipated

**Table 2. Socio-demographic characteristics of patients diagnosed with a chronic illness in Amhara Region Referral Hospitals, Ethiopia (n = 173).**

| S.no | Variable | | Frequency (%) |
|---|---|---|---|
| 1 | **Sex** | Male | 69 (39) |
| | | Female | 104 (60) |
| 2 | **Residency** | Urban | 135 (78) |
| | | Rural | 38 (22) |
| 3 | **Ethnicity** | Amhara | 165 (95) |
| | | Oromo | 3 (2) |
| | | Others | 5 (3) |
| 4 | **Religion** | Orthodox | 135 (79) |
| | | Muslim | 32 (18) |
| | | Protestant & Catholic | 6 (3) |
| 5 | **Marital status** | Single | 27 (15) |
| | | Married | 109 (63) |
| | | Separated | 13 (8) |
| | | Divorced & Widowed | 10 (14) |
| 6 | **Educational status** | Not write & read | 43 (25) |
| | | Read and write | 19 (11) |
| | | Primary (1–8) | 36 (20) |
| | | Secondary (9–10) | 32 (18) |
| | | Preparatory and technique | 21 (13) |
| | | Diploma and above | 22 (13) |
| 7 | **Occupation** | Government | 50 (29) |
| | | Non-Gov't & Private | 123 (71) |
| 8 | **Information from mass-media** | Yes | 140 (81) |
| | | No | 33 (19) |
| 9 | **Source of information from Television** | Yes | 119 (69) |
| | | No | 54 (31) |

stigma score to family and friend was 1.88 (1.02), to work colleagues 1.86 (0.89), and for Health Care Workers (HCW) 1.6 (0.8). The floor and ceiling effect for the total anticipated stigma score was 11.6% and 0% respectively (Table 4).

## Construct validity

**Content validity.** On the cognitive debriefing report, from the total of 13 participants on 12 items CIAS score, their mean (SD) was 30.1 (6). The mean (SD) score in family and friends was 11(3), working colleagues 11(3), and health care workers were 11.2 (3).

After cognitive data collection; an expert discussion was held one from psychiatry, one from chronic illness clinic nurse, and one from a clinical psychologist. Two experts were agreed on all components, whereas one expert (clinical psychologist) said "the questionnaire was nice, but I am afraid they may trigger (introduce) stigma to the people". With the team, we discussed anticipated stigma is not a daily event and it is their anticipation. If the patient is anticipated something in their mind it needs to address before goes to psychological and physical damage. Finally, we agreed the tool can be used to assess the anticipated stigma and identify factors related to their illness.

**Convergent validity.** The final model demonstrated convergent validity with the Kessler Psychological Distress scale (K10) and WHOQOL BREF for the CIASS at 0.01 level of Pearson correlation (Table 5).

**Table 3. Clinical characteristics of patients with chronic illness attending follow-up clinic in Amhara Region Referral Hospitals (n = 173).**

| S.no | Variables | | Frequency (%) |
|------|-----------|---|---------------|
| 1 | Fear of contagion in the hospital | Yes | 126 (73) |
| | | No | 47 (27) |
| 2 | Regular follow up in the clinic | Yes | 158 (91) |
| | | No | 15 (9) |
| 3 | Comorbidity | Yes | 56 (32) |
| | | No | 117 (68) |
| 4 | Number of comorbidities | 1 (one) | 44 (25) |
| | | More than one | 11 (6) |
| 5 | Diagnosis | Hypertension | 26 (15) |
| | | CHF | 9 (5) |
| | | DM | 30 (17) |
| | | Asthma & Neurologic | 5 (3) |
| | | HIV/AIDS | 34 (20) |
| | | Cancer | 20 (11) |
| | | Mental illness | 34 (20) |
| | | Others | 15 (9) |

**Key:-** CHF = Congestive Heart Failure, DM = Diabetes Mellitus, HIV/AIDS = Human Immuno-Deficiency Virus/Acquired Immunodeficiency Syndrome.

**Structural validity.** The KMO (Kaiser-Meyer-Olkin) sampling adequacy index on the data was 0.872 and bartlett's test of sphericity, p-value < 0.001. After varimax rotation, the 12 items of CIASS load on three components. The first component accounted for 53.8% of the total variation, the second for 13.8%, and the third component for 11.3% of the variation. It demonstrated that the CIASS is grouped into three domains consisting of family and friends, work colleagues, and health care workers (Table 6).

By using a structural equation model (SEM), the estimation equation yields all over model fitness with a root mean squared error of approximation (RMSEA) 0.12 (90% CI, 0.10–0.14), on baseline comparative fit index (CFI) of 0.92 and Tucker-Lewis Index (TLI) of 0.90. After this result, all over modification indices were run and gives a RMSEA 0.049 (90% CI, 0.012–0.075) CFI and TLI 0.98 and 0.99 respectively (Table 7 and Figs 1 and 2).

## Discussion

There are a dozen thousands of psychometric tests and tools claiming to measure an unknown character or hidden ability related to health. Anticipated stigma related to chronic illness

**Table 4. Internal consistency, the mean and standard deviation of dimensions of CIASS for patients with chronic illness in Amhara Region Referral Hospital (n = 173).**

| Dimensions of CIASS | Internal consistency (Cronbach-$\alpha$) | Floor/ceiling effect (%) | Mean (SD) |
|---------------------|------------------------------------------|--------------------------|-----------|
| Family and Friends | 0.93 | 30.6/0.03 | 1.88 (1.02) |
| Work colleagues | 0.91 | 0.24/0.01 | 1.86 (0.89) |
| HCW | 0.88 | 0.39/0.01 | 1.66 (0.80) |
| Total CIASS | 0.92 | 11.6/0 | 1.80 (0.75) |

**Key:-** CIASS = Chronic illness anticipated stigma scale, HCW = health care workers, SD = Standard Deviation.

**Table 5. Convergent validity of anticipated stigma scale among chronic illness patients in Amhara Regional Referral Hospitals (n = 173).**

| CIASS | Pearson correlation coefficient | |
| --- | --- | --- |
| | With QOL | With K10 |
| **Family and Friends** | -0.307** | 0.577** |
| **Work colleagues** | -0.352** | 0.449** |
| **HCW** | -0.271** | 0.284** |
| **Total CIASS** | -0.375** | 0.541** |

** Correlation is significant at the 0.01 level (2-tailed).

**Key**:- CMD = Common mental Distress, CIASS = Chronic illness anticipated stigma scale, QOL = Quality of Life, HCW = Health Care Workers.

measurement was first introduced by professor Valerie Earnshaw and her team among a variety of chronic disease patients [4]. This original English CIASS measure is a brief measure with good psychometric properties, which were confirmed in the USA, Iran's, Colombian, and Italian's validation studies [4, 18, 19, 35].

The study aimed to translate, culturally adapt the CIASS into Amharic language and investigate the psychometric properties of translating the version in patients with chronic illness in Ethiopia. The results support the reliability (i.e., internal consistency), and validity (i.e., structural, construct, and content) of the chronic illness anticipated stigma scale of the Amharic version.

Our finding demonstrated an excellent value of Cronbach's α coefficient for the reliability of the scale (0.92), and it was almost in line with the study conducted in the United States of America (USA) (Cronbach's alpha of 0.95) for the original English version tool development [4], and Iran's study (Cronbach's alpha of 0.88) for Persian version of anticipated stigma scale

**Table 6. Exploratory factor analysis of CIASS after Varimax rotation among patients with chronic illness in Amhara Region Referral Hospital (n = 173).**

| Rotated Component Matrix[a] | | | |
| --- | --- | --- | --- |
| **CIASS Questions** | **Component of CIASS** | | |
| | **Family & Friends** | **Work colleagues** | **HCWs** |
| Will think that your illness is your fault (Item 3) | 0.868 | | |
| Will not think as highly of you (Item 4) | 0.849 | | |
| Will be angry with you (Item 1) | 0.819 | | |
| Will blame you for not getting better (Item 2) | 0.818 | | |
| Will discriminate against you (Item 6) | | 0.862 | |
| Will not promote you (Item 5) | | 0.859 | |
| Will Assign a challenging project to someone else (Item 7) | | 0.852 | |
| Will think that you cannot fulfill your work responsibilities (Item 8) | | 0.742 | |
| Will blame you for not getting better (Item 11) | | | 0.838 |
| Will think that you are a bad patient. (Item 12) | | | 0.823 |
| Will give you poor care (Item 10) | | | 0.809 |
| Will be frustrated with you (Item 9) | | | 0.774 |

**N.B:-** Extraction Method: Principal Component Analysis (PCA).

Rotation Method: Varimax with Kaiser Normalization.

a. Rotation converged in 5 iterations.

Key:- CIASS = chronic illness anticipated stigma scale, HCW = health care workers, PCA = principal component analysis.

**Table 7. Structural equation model estimation overall fitness results before and after modification indices for anticipated stigma scale among selected patients in Amhara Region Referral Hospitals (n = 173).**

| Fit statistics | Value | | Description |
|---|---|---|---|
| | Before | After | |
| **Likelihood ratio** Chi $^2$ –ms (44) | 178.418 | 61.954 | Model vs. saturated |
| P > Chi $^2$ | 0.000 | 0.038 | |
| Chi $^2$-bs (66) | 1707.384 | 1707.384 | Baseline vs saturated |
| P > Chi $^2$ | 0.000 | 0.000 | |
| **Population error** RMSEA | 0.120 | 0.049 | Root mean squared error of approximation |
| 90CI, lower bound | 0.101 | 0.012 | |
| upper bound | 0.140 | 0.075 | Probability RMSEA < = 0.05 |
| P close | 0.000 | 0.508 | |
| **Information criteria** AIC | 4504.834 | 4402.369 | Akaike's information criterion |
| BIC | 4627.812 | 4547.421 | Bayesian information criterion |
| **Baseline comparison** CFI | 0.922 | 0.989 | Comparative fit index |
| TLI | 0.900 | 0.994 | Tuckler- Lewis index |
| **Size of residuals** SRMR | 0.054 | 0.034 | Standardized root mean square residual |
| CD | 0.999 | 0.998 | Coefficient of determination |

Key:- AIC = Akaike's Information Criterion, BIC = Bayesian Information Criterion, CD = Coefficient of determination, RMSEA = Root Mean Square Error of Approximation, TLI = Tuckler-Lewis Index.

[18]. The main reason for the observed consistency could be due to the use of a standard method of language translation and following each step of translation and psychometric test daily. Our finding was higher than in the study which was conducted among Colombian Spanish-speaking patients (Cronbach's alpha of 0.81) [19]. The main reason for the discrepancy could be due to our sample incorporating almost all chronic disease type patients.

Different research had shown that anticipated stigma affects the physical and mental well-being of people living with chronic illnesses [9, 13]. Therefore, we verified the convergent validity of CIASS relative to other constructs that are related to mental health and quality of life. The convergent validity translated tool was positively correlated with Kessler psychological distress scale. The possible reason for directly related correlation was due to those patients who scored a high level of psychological distress on the K10 tool may have a high level of anticipated stigma [22]. Also, it was negatively correlated with WHO QOL BREF, based on different types of evidence, those who score high levels of QOL may have a low level of anticipated stigma [23, 24]. These correlated findings were supported by the Spanish and Persian versions of CIASS validation studies [18, 19]. The possible reasons for similarity may be we followed the original English version tool developer suggestion and recommendation in the translation and cultural adaptation process.

Structural validity by using confirmatory factor analysis showed three components of anticipated stigma with the total score covers 78.8% data variance. The scale includes three factors; namely, anticipated stigma from family and friends, work colleagues, and health care workers. Our finding was similar to the Persian version anticipated stigma tool and the English version of the anticipated stigma tool [4, 18]. The tool was developed based on the concept related to anticipated stigma and it was more similar to the previous findings. However, our finding was higher than the study conducted among Colombian Spanish-speaking patients (a 3-factor structure explained 61.8% of the variance) [19]. The possible reason for the discrepancy may be in our study we incorporated all chronic illnesses, including HIV/AIDS and Mental illness. Therefore, we may fit a more homogenous population for our study.

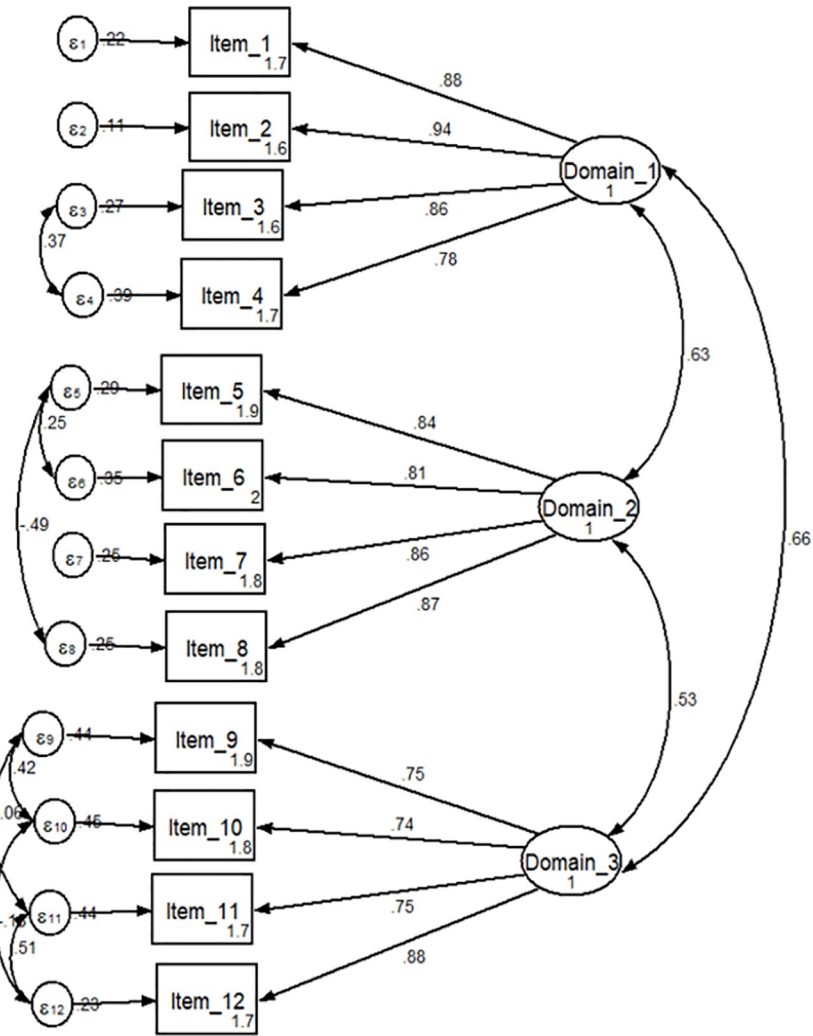

Key:- Domain_1 = family & friends, Domain_2 = Work colleagues, & Domain_3 = Health care workers (HCW), for details of each item description please see Table 6.

**Fig 1.**

This Amharic language version of chronic illness anticipated stigma shows a good model fit in RMSEA (0.049). RMSEA is one of the absolute indexes that describe closeness to fit. Values below 0.05 indicate a good model fit [36]. On other indicators of model fitness in CFI (0.98) and TLI (0.99), it was a good model fit for the study population. Based on different scholars; values above 0.95 indicate a good model fit [37–39]. Our finding was almost in line with the study conducted among Colombian and the original English version studies [4, 19]. Following the recommended method of psychometric analysis may lead to similar findings with others.

To the best of our knowledge, this is the first validation of the CIASS in the Amharic language-speaking population. Following standard procedure for the linguistic validation process, and availability of all data on Mendeley at DOI 10.17632/gvrk8njd76.1 was the strength of our study. Furthermore, the Amharic version of CIASS was evaluated among people living with a variety of chronic illnesses. Therefore, it can measure anticipated stigma among people living with a variety of chronic illnesses has an advantage.

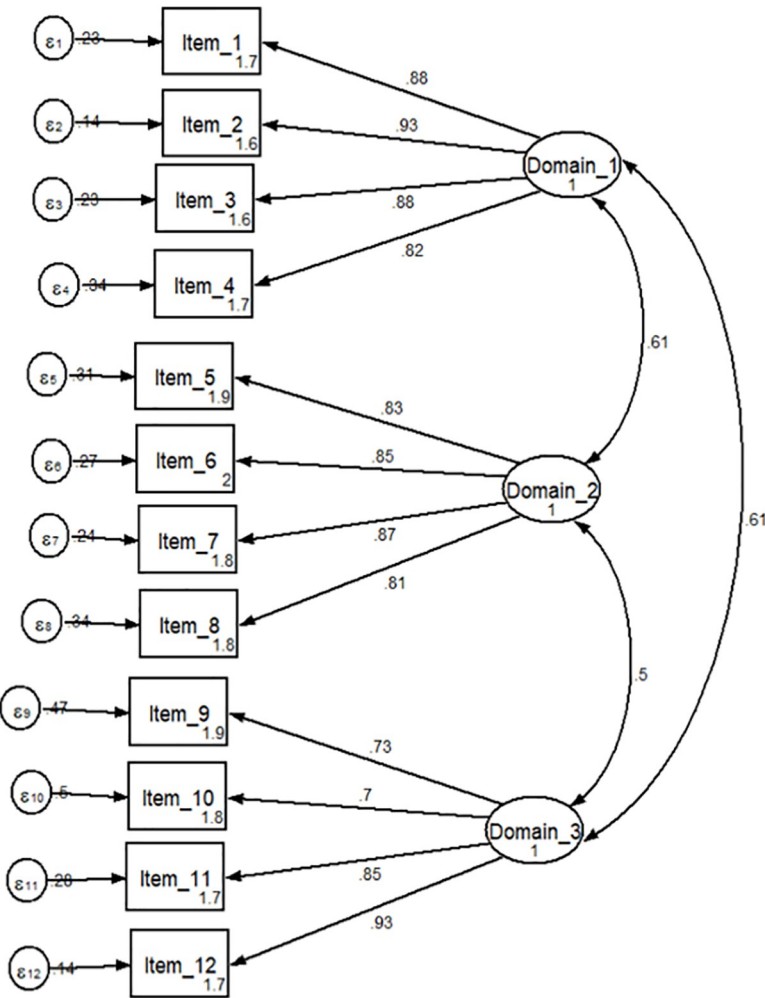

Key:- Domain_1 = family & friends, Domain_2 = Work colleagues, & Domain_3 = Health care workers (HCW), for details of each item description please see Table 6.

**Fig 2.**

There are several limitations of the CIASS that are important to consider. The psychometric properties of the Amharic Version CIASS were evaluated using samples of people living with chronic illnesses who reside in Ethiopia. The extent to which it is generalizable to other geographic locations is unknown. COVID-19 precaution makes us difficult to get patients repeatedly for test-retest reliability, and difficulty to get the same tool to compare convergent and divergent validity were the limitations.

## Conclusion and recommendation

This validation and psychometric study provide preliminary evidence of the Amharic version of the chronic illness anticipated stigma scale. This tool may provide specific information on the extent to which Amharic-speaking patients are anticipating chronic illness stigma and from whom. This may allow researchers and concerned organizations to better tailor stigma interventions to achieve maximum effectiveness.

Also, it may be important for an international and local researcher for comparing the magnitude of anticipated stigma among different population groups. Such type of research can trigger and easily support the researcher to see the level of anticipated stigma in the same type of population groups. Multiple language translation and psychometric tests should help communication between patients and physicians in different regions of Ethiopia, which could lead to more consistent assessment and management of chronic illness patients. Furthermore, a future researcher should continue to examine larger population groups by translating the original scale to the language of targeted population groups. Assessing the burden of stigma is about protecting and investigating together in good health and wellbeing is one of the sustainable development goals (SDG3) in Ethiopia.

## Supporting information

**S1 File. Questionnaire for CIASS English version.**
(PDF)

**S1 Data. Anticipated stigma tool validation main data.**
(XLSX)

## Acknowledgments

Special thanks should be given to Professor Valerie Earnshaw for her permission to translate the tool. Also, our deepest gratitude goes to all patients, data collectors, and facilitators (Mr. Alebachew G, Dr. Gashaw A, Dr. Teshome M, and Sr. Belaynesh T.) for their unreserved support throughout the data collection period.

## Author Contributions

**Conceptualization:** Mohammed Hassen Salih, Lena Wettergren, Helena Lindgren, Kerstin Erlandsson, Hussen Mekonen, Lemma Derseh.

**Data curation:** Mohammed Hassen Salih.

**Formal analysis:** Mohammed Hassen Salih, Lena Wettergren, Helena Lindgren, Kerstin Erlandsson, Hussen Mekonen, Lemma Derseh.

**Investigation:** Mohammed Hassen Salih, Lena Wettergren, Helena Lindgren, Kerstin Erlandsson, Hussen Mekonen, Lemma Derseh.

**Methodology:** Mohammed Hassen Salih, Lena Wettergren, Helena Lindgren, Kerstin Erlandsson, Hussen Mekonen, Lemma Derseh.

**Project administration:** Mohammed Hassen Salih, Lena Wettergren, Kerstin Erlandsson.

**Resources:** Kerstin Erlandsson.

**Software:** Mohammed Hassen Salih, Kerstin Erlandsson, Hussen Mekonen, Lemma Derseh.

**Supervision:** Lena Wettergren, Helena Lindgren, Kerstin Erlandsson, Hussen Mekonen, Lemma Derseh.

**Validation:** Mohammed Hassen Salih, Lena Wettergren, Helena Lindgren, Kerstin Erlandsson, Hussen Mekonen, Lemma Derseh.

**Visualization:** Mohammed Hassen Salih, Hussen Mekonen.

**Writing – original draft:** Mohammed Hassen Salih, Kerstin Erlandsson.

**Writing – review & editing:** Mohammed Hassen Salih, Lena Wettergren, Helena Lindgren, Kerstin Erlandsson, Hussen Mekonen, Lemma Derseh.

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
