## [Decision Letter · Decision Letter 0]

6 Oct 2021

PONE-D-21-24161Translation and psychometric evaluation of Chronic Illness Anticipated Stigma Scale (CIASS) among patients in EthiopiaPLOS ONE

Dear Dr. Hassen,

Thank you for submitting your manuscript to PLOS ONE. After careful consideration, we feel that it has merit but does not fully meet PLOS ONE’s publication criteria as it currently stands. Therefore, we invite you to submit a revised version of the manuscript that addresses the points raised during the review process.

In Discussion section, the authors should summarize their observations and how they were similar or different from prior studies. This reviewer would expect to see some points regarding how to translate these observations to help address this public health concern. 

We look forward to receiving your revised manuscript.

Kind regards,

Wen-Jun Tu

Academic Editor

PLOS ONE

a) Did participants provide their written or verbal informed consent to participate in this study?

b) If consent was verbal, please explain i) why written consent was not obtained, ii) how you documented participant consent, and iii) whether the ethics committees/IRB approved this consent procedure

4. Please include your full ethics statement in the ‘Methods’ section of your manuscript file. In your statement, please include the full name of the IRB or ethics committee who approved or waived your study, as well as whether or not you obtained informed written or verbal consent. If consent.

Reviewers' comments:

Reviewer's Responses to Questions

**Comments to the Author**

1. Is the manuscript technically sound, and do the data support the conclusions?

Reviewer #1: Yes

2. Has the statistical analysis been performed appropriately and rigorously? 

Reviewer #1: Yes

3. Have the authors made all data underlying the findings in their manuscript fully available?

Reviewer #1: Yes

4. Is the manuscript presented in an intelligible fashion and written in standard English?

Reviewer #1: No

5. Review Comments to the Author

Reviewer #1: Thanks for the opportunity to review this manuscript. The authors exploratorily translated the Chronic Illness Anticipated Stigma Scale (CIASS) into the Amharic language and assessed the reliability and validity of the scale in patients with chronic illness. Although, the findings do provide useful tool for scale localization. I have many concerns with the paper in its current form. I describe these concerns below, separating them into different parts.

Introduction:

1. Please adjust the structure of the Introduction. Why there are only two sentences in the first paragraph?

2. Some conclusion seems needing citations, such as: the second sentence in the 2nd paragraph.

3. The introduction is too long to introduce the backgrounds for the authors. It is not needed to have a literature review for an article, and the authors just introduced the research progress in this field.

4. Please add the aims of the current study in the last paragraph in the Introduction

Methods:

1. Some basic information was not shown in the Participants part. Please introduce the details of the inclusive and exclusive criteria. How do the authors define the chronic patients, and what chronic disorders were included?

2. Why do the authors introduce the related factors of stigma in the Methods? Such as the 2nd paragraph in Variable and Measures. Please adjust the content.

3. Why do the authors select 13 patients to do cognitive interviews?

4. Analytical process: the authors should introduce the details of the primary statistical analysis process. Moreover, have the authors run other models for factor analysis, and why do the authors choose PCA?

5. Please add the inter-rater correlation coefficient of the scale score after two-day training for the collectors.

Results:

1. Please add abbreviations for each Table and Fig.

2. Please adjust the background of the two Figs, maybe the white background is better.

3. Some contents in this part are not suitable, which are better in the Methods.

Discussion:

Discussion is too short. Many points need to explore in depth.

6. PLOS authors have the option to publish the peer review history of their article (what does this mean?). If published, this will include your full peer review and any attached files.

Reviewer #1: No

---

## [Author Response · Author response to Decision Letter 0]

29 Oct 2021

Dear the reviewer

We are so grateful for your scientific comments and suggestion.

we tried to revise most parts and respond to each valuable comment.

And attached the reviewer's comment.

Thank you so much for your precious time to review.

---

## [Editor Report · Decision Letter 1]

17 Dec 2021

PONE-D-21-24161R1Translation and psychometric evaluation of Chronic Illness Anticipated Stigma Scale (CIASS) among patients in EthiopiaPLOS ONE

Dear Dr. Hassen,

Thank you for submitting your manuscript to PLOS ONE. After careful consideration, we feel that it has merit but does not fully meet PLOS ONE’s publication criteria as it currently stands. Therefore, we invite you to submit a revised version of the manuscript that addresses the points raised during the review process.

In Discussion section, the authors should summarize their observations and how they were similar or different from prior studies. This editor would expect to see some points regarding how to translate these observations to help address this public health concern.

We look forward to receiving your revised manuscript.

Kind regards,

Wen-Jun Tu

Academic Editor

PLOS ONE
---

## [Author Response · Author response to Decision Letter 1]

4 Jan 2022

Dear Editors

Thank you for an excellent opportunity to involve in such an important scientific journal to be one of the selected manuscripts to this stage. 

Here, I will try to elaborate on each raised comment and issues

1. In the Discussion section, the authors should summarize their observations and how they were similar or different from prior studies. The editor would expect to see some points regarding how to translate these observations to help address this public health concern.

Response – Thank you for your excellent insight and important consideration in the discussion section. We tried to incorporate the new points which will add to compare and contrast our findings. Also, in the conclusion and discussion section, we tried to add points on the scientific merit. 

2. If applicable, we recommend that you deposit your laboratory protocols in protocols.io to enhance the reproducibility of your results.

Response – It was an excellent reminding, but our study doesn’t have any protocols. Except for our main data which was attached in the Mendeley Data Depository (DOI 0.17632/gvrk8njd76.1).

Journal Requirements

Response – We thank you for the comments related to the reference section. Dear editor before we submit to the journal used Zotero reference software and we ask for an excuse to our reference mix arrangement. We attached in the next Table 1 for each change and reason to change.

Table 1. List of references that were revised based on the current manuscript.

S.no Previous reference list New reference list Reasons for change

01 Ref 1 & 2 No change

02 Ref 3 Change done Revised based on the current source from free internet

03 Ref 4 up to 10 No change

04 Ref 11 Removed It was redundant with reference number 04

05 Ref 12 up to 15 Changed to 11 up to 14 Reference order based

06 Ref 16 Reference 15 Edited the author list

07 Ref 17 Reference 16 All parts edited based on the current internet availability

08 Ref 18 &19 Changed to 17 & 18 Based on reference order

09 Ref 20 (Removed) Added new ref 19 (Trejos-Herrera A. et.al) We found the recent literature which supported our paper.

10 Ref 20 up to Ref 25 No change

11 Added new ref 26 This reference supported our K10 psychological distress scale

12 Ref 26 up to ref 33 Changed to ref 27 up to ref 34 Based on reference order

13 Added new Ref 35 It was one of the supportive papers we missed in the previous literature.

14 Ref 34 up to 37 Changed to ref 36 up to Ref 39 Based on the reference order

2. While revising your submission, please upload your figure files to the Pre flight Analysis and Conversion Engine (PACE) digital diagnostic tool.

Response – thank you for your notification on the standard digital diagnostic tool. Created the account and got better picture quality. Uploaded our picture (Fig 1 & Fig 2).

We thank you for our repeated conversation and support

---

## [Editor Report · Decision Letter 2]

5 Jan 2022

Translation and psychometric evaluation of Chronic Illness Anticipated Stigma Scale (CIASS) among patients in Ethiopia

PONE-D-21-24161R2

Dear Dr. Hassen,

We’re pleased to inform you that your manuscript has been judged scientifically suitable for publication and will be formally accepted for publication once it meets all outstanding technical requirements.

Kind regards,

Wen-Jun Tu

Academic Editor

PLOS ONE
---

## [Editor Report · Acceptance letter]

12 Jan 2022

PONE-D-21-24161R2 

Translation and psychometric evaluation of chronic illness anticipated stigma scale (CIASS) among patients in Ethiopia 

Dear Dr. Salih:

I'm pleased to inform you that your manuscript has been deemed suitable for publication in PLOS ONE. Congratulations! Your manuscript is now with our production department. 

Kind regards, 

on behalf of

Dr. Wen-Jun Tu 

Academic Editor

PLOS ONE